# The Eighty Five Percent Rule for optimal learning

Robert C. Wilson[1,2]*, Amitai Shenhav[3,4], Mark Straccia[5] & Jonathan D. Cohen[6]

Researchers and educators have long wrestled with the question of how best to teach their clients be they humans, non-human animals or machines. Here, we examine the role of a single variable, the difficulty of training, on the rate of learning. In many situations we find that there is a sweet spot in which training is neither too easy nor too hard, and where learning progresses most quickly. We derive conditions for this sweet spot for a broad class of learning algorithms in the context of binary classification tasks. For all of these stochastic gradient-descent based learning algorithms, we find that the optimal error rate for training is around 15.87% or, conversely, that the optimal training accuracy is about 85%. We demonstrate the efficacy of this 'Eighty Five Percent Rule' for artificial neural networks used in AI and biologically plausible neural networks thought to describe animal learning.

[1] Department of Psychology, University of Arizona, Tucson, AZ, USA. [2] Cognitive Science Program, University of Arizona, Tucson, AZ, USA. [3] Cognitive, Linguistic, & Psychological Sciences, Brown University, Providence, RI, USA. [4] Brown Institute for Brain Science, Brown University, Providence, RI, USA. [5] Department of Psychology, UCLA, Los Angeles, CA, USA. [6] Princeton Neuroscience Institute, Princeton University, Princeton, NJ, USA. *email: bob@arizona.edu

When we learn something new, like a language or musical instrument, we often seek challenges at the edge of our competence—not so hard that we are discouraged, but not so easy that we get bored. This simple intuition, that there is a sweet spot of difficulty, a 'Goldilocks zone'[1], for motivation and learning is at the heart of modern teaching methods[2] and is thought to account for differences in infant attention between more and less learnable stimuli[1]. In the animal learning literature it is the intuition behind shaping[3] and fading[4], whereby complex tasks are taught by steadily increasing the difficulty of a training task. It is also observable in the nearly universal 'levels' feature in video games, in which the player is encouraged, or even forced, to a higher level of difficulty once a performance criterion has been achieved. Similarly in machine learning, steadily increasing the difficulty of training has proven useful for teaching large scale neural networks in a variety of tasks[5,6], where it is known as 'Curriculum Learning'[7] and 'Self-Paced Learning'[8].

Despite this long history of empirical results, it is unclear why a particular difficulty level may be beneficial for learning nor what that optimal level might be. In this paper we address this issue of optimal training difficulty for a broad class of learning algorithms in the context of binary classification tasks, in which ambiguous stimuli must be classified into one of two classes (e.g., cat or dog).

In particular, we focus on the class of stochastic gradient-descent based learning algorithms. In these algorithms, parameters of the model (e.g., the weights in a neural network) are adjusted based on feedback in such a way as to reduce the average error rate over time[9]. That is, these algorithms descend the gradient of error rate as a function of model parameters. Such gradient-descent learning forms the basis of many algorithms in AI, from single-layer perceptrons to deep neural networks[10], and provides a quantitative description of human and animal learning in a variety of situations, from perception[11], to motor control[12] to reinforcement learning[13]. For these algorithms, we provide a general result for the optimal difficulty in terms of a target error rate for training. Under the assumption of a Gaussian noise

process underlying the errors, this optimal error rate is around 15.87%, a number that varies slightly depending on the noise in the learning process. That is the optimal accuracy for training is around 85%. We show theoretically that training at this optimal difficulty can lead to exponential improvements in the rate of learning. Finally, we demonstrate the applicability of the Eighty Five Percent Rule to artificial one- and two-layer neural networks[9,14], and a model from computational neuroscience that is thought to describe human and animal perceptual learning[11].

## Results

**Optimal training difficulty for binary classification tasks.** In a standard binary classification task, an animal or machine 'agent' makes binary decisions about simple stimuli. For example, in the classic Random Dot Motion paradigm from Psychology and Neuroscience[15,16], stimuli consist of a patch of moving dots—most moving randomly but a small fraction moving coherently either to the left or the right—and participants must decide in which direction the coherent dots are moving. A major factor in determining the difficulty of this perceptual decision is the fraction of coherently moving dots, which can be manipulated by the experimenter to achieve a fixed error rate during training using a procedure known as 'staircasing'[17].

We assume that agents make their decision on the basis of a scalar, subjective decision variable, $h$, which is computed from a stimulus that can be represented as a vector **x** (e.g., the direction of motion of all dots)

$$h = \Phi(\mathbf{x}, \boldsymbol{\phi}) \quad (1)$$

where $\Phi(\cdot)$ is a function of the stimulus and (tunable) parameters $\boldsymbol{\phi}$. We assume that this transformation of stimulus $x$ into the subjective decision variable $h$ yields a noisy representation of the true decision variable, $\Delta$ (e.g., the fraction of dots moving left). That is, we write

$$h = \Delta + n \quad (2)$$

where the noise, $n$, arises due to the imperfect representation of the decision variable. We further assume that this noise, $n$, is random and sampled from a zero-mean Gaussian distribution with standard deviation $\sigma$ (Fig. 1a).

If the decision boundary is set to 0, such that the model chooses option A when $h > 0$, option B when $h < 0$ and randomly when $h = 0$, then the noise in the representation of the decision variable leads to errors with probability

$$\mathrm{ER} = \int_{-\infty}^{0} p(h|\Delta, \sigma)\mathrm{d}h = \mathrm{F}(-\Delta/\sigma) = \mathrm{F}(-\beta\Delta) \quad (3)$$

where $F(x)$ is the cumulative density function of the standardized noise distribution, $p(x) = p(x|0, 1)$, and $\beta = 1/\sigma$ quantifies the precision of the representation of $\Delta$ and the agent's skill at the task. As shown in Fig. 1b, this error rate decreases as the decision gets easier ($\Delta$ increases) and as the agent becomes more accomplished at the task ($\beta$ increases).

The goal of learning is to tune the parameters $\boldsymbol{\phi}$ such that the subjective decision variable, $h$, is a better reflection of the true decision variable, $\Delta$. That is, the model should aim to adjust the parameters $\boldsymbol{\phi}$ so as to decrease the magnitude of the noise $\sigma$ or, equivalently, increase the precision $\beta$. One way to achieve this tuning is to adjust the parameters using gradient descent on the error rate, i.e. changing the parameters over time $t$ according to

$$\frac{\mathrm{d}\boldsymbol{\phi}}{\mathrm{d}t} = -\eta \nabla_{\boldsymbol{\phi}} \mathrm{ER} \quad (4)$$

where $\eta$ is the learning rate and $\nabla_{\boldsymbol{\phi}}\mathrm{ER}$ is the derivative of the error rate with respect to parameters $\boldsymbol{\phi}$. This gradient can be

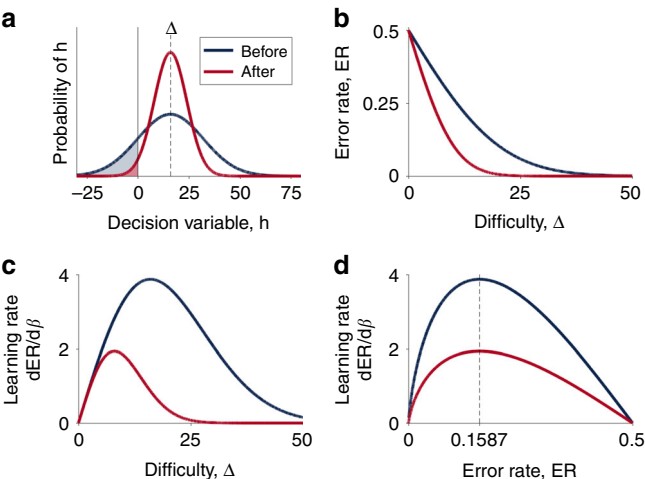

**Fig. 1** Illustration of the model. **a** Distributions over decision variable $h$ given a particular difficulty, $\Delta = 16$, with lower precision before learning and higher precision after learning. The shaded regions corresponds to the error rate—the probability of making an incorrect response at each difficulty. **b** The error rate as a function of difficulty before and after learning. **c** The derivative that determines the rate of learning as a function of difficulty before and after learning showing that the optimal difficulty for learning is lower after learning than before. **d** The same derivative as in **c** re-plotted as a function of error rate showing that the optimal error rate (at 15.87% or ~85% accuracy) is the same both before and after learning

written in terms of the precision, $\beta$, as

$$\nabla_\phi \text{ER} = \frac{\partial \text{ER}}{\partial \beta} \nabla_\phi \beta \qquad (5)$$

Note here that only the first term on the right hand side of Eq. (5) depends on the difficulty $\Delta$, while the second describes how the precision changes with $\phi$. Note also that $\Delta$ itself, as the 'true' decision variable, is independent of $\phi$. This means that the optimal difficulty for training, that maximizes the change in the parameters, $\phi$, *at this time point*, is the value of the decision variable $\Delta^*$ that maximizes $\partial \text{ER}/\partial \beta$. Of course, this analysis ignores the effect of changing $\phi$ on the *form* of the noise—instead assuming that it only changes the scale factor, $\beta$, an assumption that likely holds in the relatively simple cases we consider here, although whether it holds in more complex cases will be an important question for future work.

In terms of the decision variable, the optimal difficulty changes as a function of precision (Fig. 1c) meaning that the difficulty of training must be adjusted online according to the skill of the agent. Using the monotonic relationship between $\Delta$ and ER (Fig. 1b) it is possible to express the optimal difficulty in terms of the error rate, $\text{ER}^*$ (Fig. 1d). Expressed this way, the optimal difficulty is constant as a function of precision, meaning that optimal learning can be achieved by clamping the error rate during training at a fixed value, which, for Gaussian noise is

$$\text{ER}^* = \frac{1}{2}\left(1 - \text{erf}\left(\frac{1}{\sqrt{2}}\right)\right) \approx 0.1587 \qquad (6)$$

That is, the optimal error rate for learning is 15.87%, and the optimal accuracy is around 85%. We call this the Eighty Five Percent Rule for optimal learning.

**Dynamics of learning**. While the previous analysis allows us to calculate the error rate that maximizes the rate of learning, it does not tell us how much faster learning occurs at this optimal error rate. In this section we address this question by comparing learning at the optimal error rate with learning at a fixed error rate, $\text{ER}_f$ (which may be suboptimal), and, alternatively, a fixed difficulty, $\Delta_f$. If stimuli are presented one at a time (i.e., *not* batch learning), in both cases, gradient-descent based updating of the parameters, $\phi$, (Eq. (4)) implies that the precision $\beta$ evolves in a similar manner, i.e..

$$\frac{d\beta}{dt} = -\eta \frac{\partial \text{ER}}{\partial \beta} \qquad (7)$$

For fixed error rate, $\text{ER}_f$, as shown in the Methods, integrating Eq. (7) gives

$$\beta(t) = \sqrt{\beta_0^2 + 2\eta K_f(t - t_0)} \qquad (8)$$

where $t_0$ is the initial time point, $\beta_0$ is the initial value of $\beta$ and $K_f$ is the following function of the training error rate

$$K_f = -F^{-1}(\text{ER}_f)p(F^{-1}(\text{ER}_f)) \qquad (9)$$

Thus, for fixed training error rate the precision grows as the square root of time with the exact rate determined by $K_f$ which depends on both the training error rate and the noise distribution.

For fixed decision variable, $\Delta_f$, integrating Eq. (7) is more difficult and the solution depends more strongly on the distribution of the noise. In the case of Gaussian noise, there is no closed form solution for $\beta$. However, as shown in the Methods, an approximate form can be derived at long times where we find that $\beta$ grows as

$$\beta(t) \propto \sqrt{\log t} \qquad (10)$$

i.e., exponentially slower than Eq. (38).

**Simulations**. To demonstrate the applicability of the Eighty Five Percent Rule we simulated the effect of training accuracy on learning in three cases, two from AI and one from computational neuroscience. From AI we consider how training at 85% accuracy impacts learning in the the simple case of a one-layer Perceptron[14] with artificial stimuli, and in the more complex case of a two-layer neural network[9] with stimuli drawn from the MNIST (Modified National Institute of Standards and Technology) dataset of handwritten digits[18]. From computational neuroscience we consider the model of Law and Gold[11], that accounts for both the behavior and neural firing properties of monkeys learning the Random Dot Motion task. In all cases we see that learning is maximized when training occurs at 85% accuracy.

**Perceptron with artificial stimuli**. The Perceptron is a classic one-layer neural network model that learns to map multi-dimensional stimuli $\mathbf{x}$ onto binary labels, $y$ via a linear threshold process[14]. To implement this mapping, the Perceptron first computes the decision variable $h$ as

$$h = \mathbf{w} \cdot \mathbf{x} \qquad (11)$$

where $\mathbf{w}$ are the weights of the network, and then assigns the label according to

$$y = \begin{cases} 1 & h > 0 \\ 0 & h \le 0 \end{cases} \qquad (12)$$

The weights, $\mathbf{w}$, which constitute the parameters of the model, are updated based on feedback about the true label $t$ by a the learning rule,

$$\mathbf{w} \leftarrow \mathbf{w} + (t - y)\mathbf{x} \qquad (13)$$

This learning rule implies that the Perceptron only updates its weights when the predicted label $y$ does not match the actual label $t$—that is, the Perceptron only learns when it makes mistakes. Naively then, one might expect that optimal learning would involve maximizing the error rate. However, because Eq. (13) is closely related (albeit not identical) to a gradient descent based rule (e.g., Chapter 39 in ref. [19]), the analysis of the previous sections applies and the optimal error rate for training is 15.87%.

To test this prediction we simulated the Perceptron learning rule for a range of training error rates between 0.01 and 0.5 in steps of 0.01 (1000 simulations per error rate, 1000 trials per simulation). Error rate was kept constant by varying the difficulty, and the degree of learning was captured by the precision $\beta$ (see Methods). As predicted by the theory, the network learns most effectively when trained at the optimal error rate (Fig. 2a) and the dynamics of learning are well described, up to a scale factor, by Eq. (38) (Fig. 2b).

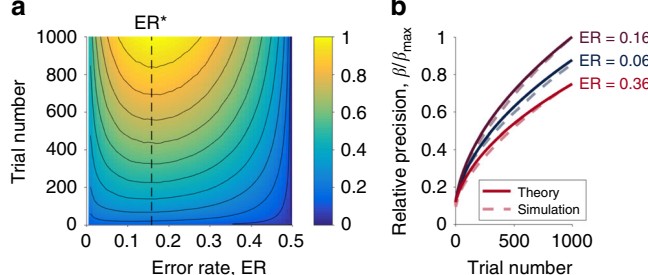

**Fig. 2** The Eighty Five Percent Rule applied to the Perceptron. **a** The relative precision, $\beta/\beta_{max}$, as a function of training error rate and training duration. Training at the optimal error rate leads to the fastest learning throughout. **b** The dynamics of learning agree well with the theory

**Two-layer network with MNIST stimuli**. As a more demanding test of the Eighty Five Percent Rule, we consider the case of a two-layer neural network applied to more realistic stimuli from the Modified National Institute of Standards and Technology (MNIST) dataset of handwritten digits[18]. The MNIST dataset is a labeled dataset of 70,000 images of handwritten digits (0 through 9) that has been widely used as a test of image classification algorithms (see ref. [20] for a list). The dataset is broken down into a training set consistent of 60,000 images and a test set of 10,000 images. To create binary classification tasks based on these images, we trained the network to classify the images according to either the parity (odd or even) or magnitude (less than 5 or not) of the number.

The network itself consisted of 1 input layer, with 400 units corresponding to the pixel values in the images, 1 hidden layer, with 50 neurons, and one output unit. Unlike the Perceptron, activity of the output unit was graded and was determined by a sigmoid function of the decision variable, $h$

$$y = \frac{1}{1 + \exp(h)} = S(h) \tag{14}$$

where the decision variable was given by

$$h = \mathbf{w}_2 \cdot \mathbf{a} \tag{15}$$

where $\mathbf{w}_2$ were the weights connecting the hidden layer to the output units and $\mathbf{a}$ was the activity in the hidden layer. This hidden-layer activity was also determined by a sigmoidal function

$$\mathbf{a} = S(\mathbf{w}_1 \cdot \mathbf{x}) \tag{16}$$

where the inputs, $\mathbf{x}$, corresponds to the pixel values in the image and $\mathbf{w}_1$ were the weights from the input layer to the hidden layer.

All weights were trained using the Backpropagation algorithm[9] which takes the error,

$$e = t - y \tag{17}$$

and propagates it backwards through the network, from output to input stage, as a teaching signal for the weights. This algorithm implements stochastic gradient descent and, if our assumptions are met, should optimize learning at a training accuracy of 85%.

To test this prediction we trained the two-layer network for 5000 trials to perform either the Parity or the Magnitude Task while clamping the training error rate between 5 and 30% (Fig. 3). After training, performance was assessed on the entire test set and the whole process was repeated 1000 times for each task. As shown in Fig. 3, training error rate has a relatively large effect on test accuracy, around 10% between the best and worse training accuracies. Moreover, for both tasks, the optimal training occurs

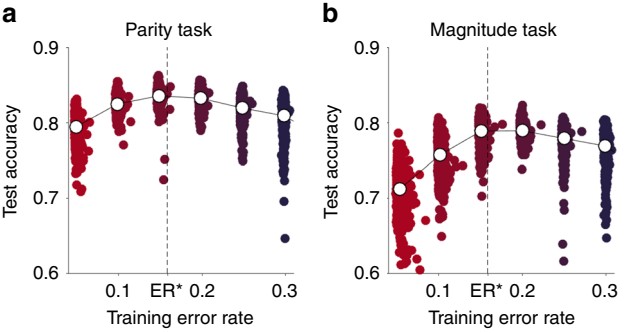

**Fig. 3** The Eighty Five Percent Rule applied to a multilayered neural network. Test accuracy vs training error rate on the MNIST dataset for the **a** Parity and **b** Magnitude tasks for 1000 different simulations. In both cases the test accuracy peaks at or near the optimal error rate. Each color corresponds to a different target training accuracy

at 85% training accuracy. This suggests that the 85% rule holds even for learning of more realistic stimuli, by more complex multi-layered networks.

**Biologically plausible model of perceptual learning**. To demonstrate how the Eighty Five Percent Rule might apply to learning in biological systems, we simulated the Law and Gold model of perceptual learning[11]. This model has been shown to capture the long term changes in behavior, neural firing and synaptic weights as monkeys learn to perform the Random Dot Motion task.

Specifically, the model assumes that monkeys make the perceptual decision between left and right on the basis of neural activity in area MT—an area in the dorsal visual stream that is known to represent motion information[15]. In the Random Dot Motion task, neurons in MT have been found to respond to both the direction $\theta$ and coherence COH of the dot motion stimulus such that each neuron responds most strongly to a particular 'preferred' direction and that the magnitude of this response increases with coherence. This pattern of firing is well described by a simple set of equations (see "Methods") and thus the noisy population response, $\mathbf{x}$, to a stimulus of arbitrary direction and coherence is easily simulated.

From this MT population response, Law and Gold proposed that animals construct a decision variable in a separate area of the brain (lateral interparietal area, LIP) as the weighted sum of activity in MT; i.e.,

$$h = \mathbf{w} \cdot \mathbf{x} + \epsilon \tag{18}$$

where $\mathbf{w}$ are the weights between MT and LIP neurons and $\epsilon$ is random neuronal noise that cannot be reduced by learning. The presence of this irreducible neural noise is a key difference between the Law and Gold model (Eq. 18) and the Perceptron (Eq. 11) as it means that no amount of learning can lead to perfect performance. However, as shown in the Methods section, the presence of irreducible noise does not change the optimal accuracy for learning which is still 85%.

Another difference between the Perceptron and the Law and Gold model is the form of the learning rule. In particular, weights are updated according to a reinforcement learning rule based on a reward prediction error

$$\delta = r - E[r] \tag{19}$$

where $r$ is the reward presented on the current trial (1 for a correct answer, 0 for an incorrect answer) and $E[r]$ is the predicted reward

$$E[r] = \frac{1}{1 + \exp(-B|h|)} \tag{20}$$

where $B$ is a proportionality constant that is estimated online by the model (see "Methods"). Given the prediction error, the model updates its weights according to

$$\mathbf{w} \leftarrow \mathbf{w} + \eta C \delta \mathbf{x} \tag{21}$$

where $C$ is the choice ($-1$ for left, $+1$ for right) and $\eta$ is the learning rate. Despite the superficial differences with the Perceptron learning rule (Eq. (13)) the Law and Gold model still implements stochastic gradient descent on the error rate[13] and learning should be optimized at 85%.

To test this prediction we simulated the model at a variety of different target training error rates. Each target training rate was simulated 100 times with different parameters for the MT neurons (see "Methods"). The precision, $\beta$, of the trained network was estimated by fitting simulated behavior of the network on a set of test coherences that varied logarithmically between 1 and 100%. As shown in Fig. 4a the precision after training is well

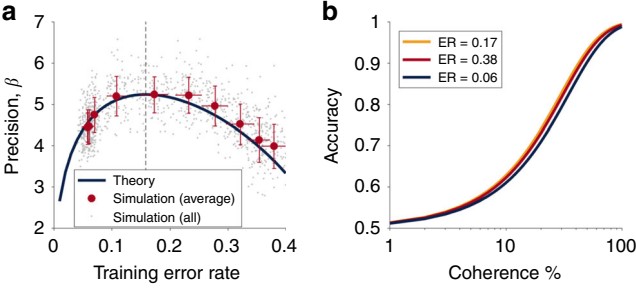

**Fig. 4** The Eighty Five Percent Rule applied to the Law and Gold model of perceptual learning. **a** Precision of the trained network as function of training error rate. Gray dots represent the results of individual simulations —note that the empirical error rate on each run often differs slightly from the target error rate due to noise. Red dots correspond to the average precision and empirical error rate for each target error rate (error bars ± standard deviation in both measures). **b** Accuracy as a function of coherence for the network trained at three different error rates corresponding to near optimal (ER = 0.17), too high (ER = 0.38) and too low (ER = 0.06)

described (up to a scale factor) by the theory. In addition, in Fig. 4b, we show the expected difference in behavior—in terms of psychometric choice curves—for three different training error rates. While these differences are small, they are large enough that they could be distinguished experimentally.

## Discussion
In this article we considered the effect of training accuracy on learning in the case of binary classification tasks and stochastic gradient-descent-based learning rules. We found that the rate of learning is maximized when the difficulty of training is adjusted to keep the training accuracy at around 85%. We showed that training at the optimal accuracy proceeds exponentially faster than training at a fixed difficulty. Finally we demonstrated the efficacy of the Eighty Five Percent Rule in the case of artificial and biologically plausible neural networks.

Our results have implications for a number of fields. Perhaps most directly, our findings move towards a theory for identifying the optimal environmental settings in order to maximize the rate of gradient-based learning. Thus the Eighty Five Percent Rule should hold for a wide range of machine learning algorithms including multilayered feedforward and recurrent neural networks (e.g. including 'deep learning' networks using back-propagation[9], reservoir computing networks[21,22], as well as Perceptrons). Of course, in these more complex situations, our assumptions may not always be met. For example, as shown in the Methods, relaxing the assumption that the noise is Gaussian leads to changes in the optimal training accuracy: from 85% for Gaussian, to 82% for Laplacian noise, to 75% for Cauchy noise (Eq. (31) in the "Methods").

More generally, extensions to this work should consider how batch-based training changes the optimal accuracy, and how the Eighty Five Percent Rule changes when there are more than two categories. In batch learning, the optimal difficulty to select for the examples in each batch will likely depend on the rate of learning relative to the size of the batch. If learning is slow, then selecting examples in a batch that satisfy the 85% rule may work, but if learning is fast, then mixing in more difficult examples may be best. For multiple categories, it is likely possible to perform similar analyses, although the mapping between decision variable and categories will be more complex as will be the error rates which could be category specific (e.g., misclassifying category 1 as category 2 instead of category 3).

In Psychology and Cognitive Science, the Eighty Five Percent Rule accords with the informal intuition of many experimentalists that participant engagement is often maximized when tasks are neither too easy nor too hard. Indeed it is notable that staircasing procedures (that aim to titrate task difficulty so that error rate is fixed during learning) are commonly designed to produce about 80–85% accuracy[17]. Similarly, when given a free choice about the difficulty of task they can perform, participants will spontaneously choose tasks of intermediate difficulty levels as they learn[23]. Despite the prevalence of this intuition, to the best of our knowledge no formal theoretical work has addressed the effect of training accuracy on learning, a test of which is an important direction for future work.

More generally, our work closely relates to the Region of Proximal Learning and Desirable Difficulty frameworks in education[24–26] and Curriculum Learning and Self-Paced Learning[7,8] in computer science. These related, but distinct, frameworks propose that people and machines should learn best when training tasks involve just the right amount of difficulty. In the Desirable Difficulties framework, the difficulty in the task must be of a 'desirable' kind, such as spacing practice over time, that promotes learning as opposed to an undesirable kind that does not. In the Region of Proximal Learning framework, which builds on early work by Piaget[27] and Vygotsky[28], this optimal difficulty is in a region of difficulty just beyond the person's current ability. Curriculum and Self-Paced Learning in computer science build on similar intuitions, that machines should learn best when training examples are presented in order from easy to hard. In practice, the optimal difficulty in all of these domains is determined empirically and is often dependent on many factors[29]. In this context, our work offers a way of deriving the desired difficulty and the region of proximal learning in the special case of binary classification tasks for which stochastic gradient-descent learning rules apply. As such our work represents the first step towards a more mathematical instantiation of these theories, although it remains to be generalized to a broader class of circumstances, such as multi-choice tasks and different learning algorithms.

With regard to different learning algorithms, it is important to note that not all models will exhibit a sweet spot of difficulty for learning. As an example, consider how a Bayesian learner with a perfect memory would infer parameters $\phi$ by computing the posterior distribution given past stimuli, $\mathbf{x}_{1:t}$, and labels, $y_{1:t}$,

$$p(\phi|\mathbf{x}_{1:t}, y_{1:t}) \propto p(y_{1:t}|\phi, \mathbf{x}_{1:t})p(\phi)$$
$$= \prod_{i=1}^{t} p(y_i|\phi, \mathbf{x}_i)p(\phi) \qquad (22)$$

where the last line holds when the label depends only on the current stimulus. Clearly this posterior distribution over parameters is independent of the ordering of the trials meaning that a Bayesian learner (with perfect memory) would learn equally well if hard or easy examples are presented first. This is not to say that Bayesian learners cannot benefit from carefully constructed training sets, but that for a given set of training items the order of presentation has no bearing on what is ultimately learned. This contrasts markedly with gradient-based algorithms, many of which try to approximate the maximum a posteriori solution of a Bayesian model, whose training is order dependent and whose learning is optimized with $\partial ER/\partial \beta$.

Finally, we note that our analysis for maximizing the gradient, $\partial ER/\partial \beta$, not only applies to learning but to any process that affects the precision of neural representations, such as attention, engagement, or more generally cognitive control[30,31]. For example, attention is known to improve the precision with which sensory stimuli are represented in the brain, e.g., ref. [32]. If

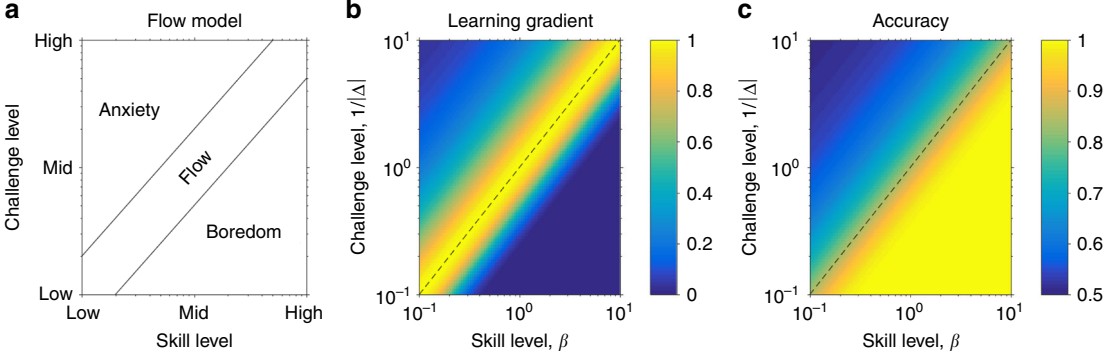

**Fig. 5** Proposed relationship between the Eighty Five Percent Rule and Flow. **a** Original model of flow as a state that is achieved when skill and challenge are well balanced. Normalized learning rate, $\partial\text{ER}/\partial\beta$, **b** and accuracy **c** as a function of skill and challenge suggests that flow corresponds to high learning and accuracy, boredom corresponds to low learning and high accuracy, while anxiety is associated with low learning and low accuracy

exerting control leads to a change in precision of $\delta\beta$, then the change in error rate associated with exerting this control is

$$\delta\text{ER} = \frac{\partial\text{ER}}{\partial\beta}\delta\beta \tag{23}$$

This predicts that the benefits of engaging cognitive control should be maximized when $\partial\text{ER}/\partial\beta$ is maximized, that is at $\text{ER}^*$. More generally this relates to the Expected Value of Control theory[30,31,33] which suggests that the learning gradient, $\partial\text{ER}/\partial\beta$, is monitored by control-related areas of the brain such as anterior cingulate cortex.

Along similar lines, our work points to a mathematical theory of the state of 'Flow'[34]. This state, 'in which an individual is completely immersed in an activity without reflective self-consciousness but with a deep sense of control' [ref. [35], p. 1], is thought to occur most often when the demands of the task are well matched to the skills of the participant. This idea of balance between skill and challenge was captured originally with a simple conceptual diagram (Fig. 5) with two other states: 'anxiety' when challenge exceeds skill and 'boredom' when skill exceeds challenge. These three qualitatively different regions (flow, anxiety, and boredom) arise naturally in our model. Identifying the precision, $\beta$, with the level of skill and the level challenge with the inverse of true decision variable, $1/\Delta$, we see that when challenge equals skill, flow is associated with a high learning rate and accuracy, anxiety with low learning rate and accuracy and boredom with high accuracy but low learning rate (Fig. 5b, c). Intriguingly, recent work by Vuorre and Metcalfe, has found that subjective feelings of Flow peaks on tasks that are subjectively rated as being of intermediate difficulty[36]. In addition work on learning to control brain computer interfaces finds that subjective, self-reported measures of 'optimal difficulty', peak at a difficulty associated with maximal learning, and not at a difficulty associated with optimal decoding of neural activity[37]. Going forward, it will be interesting to test whether these subjective measures of engagement peak at the point of maximal learning gradient, which for binary classification tasks is 85%.

## Methods

**Optimal error rate for learning.** In order to compute the optimal difficulty for training, we need to find the value of $\Delta$ that maximizes the learning gradient, $\partial\text{ER}/\partial\beta$. From Eq. (3) we have

$$\frac{\partial\text{ER}}{\partial\beta} = \Delta p(-\beta\Delta) \tag{24}$$

From here the optimal difficulty, $\Delta^*$, can be found by computing the derivative of

the gradient with respect to $\Delta$, i.e.,

$$\begin{aligned}\frac{\partial}{\partial\Delta}\frac{\partial\text{ER}}{\partial\beta} &= -\frac{\partial}{\partial\Delta}(\Delta p(-\beta\Delta)) \\ &= -p(-\beta\Delta) + \beta\Delta\frac{\partial p(x)}{\partial x}\Big|_{x=-\beta\Delta}\end{aligned} \tag{25}$$

Setting this derivative equal to zero gives us the following expression for the optimal difficulty, $\Delta^*$, and error rate, $\text{ER}^*$

$$\beta\Delta^* = \frac{p(-\beta\Delta^*)}{p'(-\beta\Delta^*)} \quad \text{and} \quad \text{ER}^* = F(-\beta\Delta^*) \tag{26}$$

where $p'(x)$ denotes the derivative of $p(x)$ with respect to $x$. Because $\beta$ and $\Delta^*$ only ever appear together in these expressions, Eq. (26) implies that $\beta\Delta^*$ is a constant. Thus, while the optimal difficulty, $\Delta^*$, changes as a function of precision (Fig. 1c), the optimal training error rate, $\text{ER}^*$ does not (Fig. 1d). That is, training with the error rate clamped at $\text{ER}^*$ is guaranteed to maximize the rate of learning.

The exact value of $\text{ER}^*$ depends on the distribution of noise, $n$, in Eq. (2). In the case of Gaussian noise, we have

$$p(x) = \frac{1}{\sqrt{2\pi}}\exp\left(-\frac{x^2}{2}\right) \tag{27}$$

which implies that

$$\frac{p(x)}{p'(x)} = -\frac{1}{x} \tag{28}$$

and that the optimal difficulty is

$$\Delta^* = \beta^{-1} \tag{29}$$

Consequently the optimal error rate for Gaussian noise is

$$\text{ER}^* = \frac{1}{2}\left(1 - \text{erf}\left(\frac{1}{\sqrt{2}}\right)\right) \approx 0.1587 \tag{30}$$

Similarly for Laplacian noise ($p(x) = \frac{1}{2}\exp(-|x|)$) and Cauchy noise ($p(x) = (\pi(1+x^2))^{-1}$) we have optimal error rates of

$$\begin{aligned}\text{ER}^*_{\text{Laplace}} &= \frac{1}{2}\exp(-1) \approx 0.1839 \\ \text{ER}^*_{\text{Cauchy}} &= \frac{1}{\pi}\arctan(-1) + \frac{1}{2} = 0.25\end{aligned} \tag{31}$$

**Optimal learning with endogenous noise.** The above analyses for optimal training accuracy also applies in the case where the decision variable, $h$, is corrupted by endogenous, irreducible noise, $\epsilon$, in addition to representation noise, $n$, that can be reduced by learning; i.e.,

$$h = \Delta + n + \epsilon \tag{32}$$

In this case we can split the overall precision, $\beta$, into two components, one based on representational uncertainty that can be reduced, $\beta_n$, and another based on endogenous uncertainty that cannot, $\beta_\epsilon$. For Gaussian noise, these precisions are related to each other by

$$\frac{1}{\beta^2} = \frac{1}{\beta_n^2} + \frac{1}{\beta_\epsilon^2} \tag{33}$$

More generally, the precisions are related by some function, $G$, such that $\beta = G(\beta_n, \beta_\epsilon)$. Since only $n$ can be reduced by learning, it makes sense to perform stochastic

gradient descent on $\beta_n$ such that the learning rule should be

$$\frac{d\beta_n}{dt} = -\eta \frac{\partial \text{ER}}{\partial \beta_n}$$
$$= -\eta \frac{\partial \text{ER}}{\partial \beta} \frac{\partial \beta}{\partial \beta_n} \quad (34)$$

Note that $\partial \beta / \partial \beta_n$ is independent of $\Delta$ so maximizing learning rate w.r.t. $\Delta$ means maximizing $\partial \text{ER} / \partial \beta$ as before. This implies that the optimal training difficulty will be the same, e.g., 85% for Gaussian noise, regardless whether endogenous noise is present or not.

**Dynamics of learning**. To calculate the dynamics of learning we need to integrate Eq. (7) over time. This, of course depends on the learning gradient, $\partial \text{ER} / \partial \beta$, which varies depending on the noise and whether the error rate or the true decision variable is fixed during training.

In the fixed error rate case, we fix the error rate during training to $\text{ER}_f$. This implies that the difficulty should change over time according to

$$\Delta(t) = -\frac{1}{\beta(t)} F^{-1}(\text{ER}_f) \quad (35)$$

where $F^{-1}(\cdot)$ is the inverse cdf. This implies that $\beta$ evolves over time according to

$$\frac{d\beta}{dt} = -\eta \frac{\partial \text{ER}}{\partial \beta}$$
$$= \eta \Delta(t) p(-\beta \Delta(t))$$
$$= -\frac{\eta}{\beta(t)} F^{-1}(\text{ER}_f) p(F^{-1}(\text{ER}_f))$$
$$= \frac{\eta K_f}{\beta(t)} \quad (36)$$

where we have introduced $K_f$ as

$$K_f = -F^{-1}(\text{ER}_f) p(F^{-1}(\text{ER}_f)) \quad (37)$$

Integrating Eq. (36) and solving for $\beta(t)$ we get

$$\beta(t) = \sqrt{\beta_0^2 + 2\eta K_f(t - t_0)} \quad (38)$$

where $t_0$ is the initial time point, and $\beta_0$ is the initial value of $\beta$. Thus, for fixed error rate the precision grows as the square root of time with the rate determined by $K_f$ which depends on both the training error rate and the noise distribution. For the optimal error rate we have, $K_f = p(-1)$.

In the fixed decision variable case, the true decision variable is fixed at $\Delta_f$ and the error rate varies as a function of time. In this case we have

$$\frac{d\beta}{dt} = -\eta \frac{\partial \text{ER}}{\partial \beta} = \Delta_f p(-\beta \Delta_f) \quad (39)$$

Formally, this can be solved as

$$\int_{\beta_0}^{\beta} \frac{1}{p(-\beta \Delta_f)} d\beta = \Delta_f(t - t_0) \quad (40)$$

However, the exact form for $\beta(t)$ will depend on $p(x)$.

In the Gaussian case we cannot derive a closed form expression for $\beta(t)$. The closest we can get is to write

$$\int_0^{\frac{\beta \Delta_f}{\sqrt{2}}} \exp(x^2) dx = \int_0^{\frac{\beta_0 \Delta_f}{\sqrt{2}}} \exp(x^2) dx + \frac{\Delta^2}{2\sqrt{\pi}}(t - t_0) \quad (41)$$

For long times, and large $\beta$, we can write

$$\int_0^{\frac{\beta \Delta_f}{\sqrt{2}}} \exp(x^2) dx < \exp\left(\frac{\beta^2 \Delta_f^2}{2}\right) \quad (42)$$

which implies that for long times $\beta$ grows slower than $\sqrt{\log t}$, which is exponentially slower than the fixed error rate case.

In contrast to the Gaussian case, the Laplacian case lends itself to closed form analysis and we can derive the following expression for $\beta$

$$\beta = \frac{1}{\Delta_f} \log\left(\exp(\beta_0 \Delta_f) + \frac{1}{2} \eta \Delta_f^2(t - t_0)\right) \quad (43)$$

Again this shows logarithmic dependence on $t$ indicating that learning is much slower with a fixed difficulty.

In the case of Cauchy noise we can compute the integral in Eq. (40) and find that $\beta$ is the root of the following equation

$$\frac{\Delta_f}{3} \beta^3 + \beta = \frac{\Delta_f}{3} \beta_0^3 + \beta_0 + \frac{\Delta_f}{\pi}(t - t_0) \quad (44)$$

For long training times this implies that $\beta$ grows as the cube root of $t$. Thus in the Cauchy case, while the rate of learning is still greatest at the optimal difficulty, the improvement is not as dramatic as in the other cases.

**Application to the perceptron**. To implement the Perceptron example, we assumed that true labels $t$ were generated by a 'Teacher Perceptron'[38] with normalized weight vector, $\mathbf{e}$. Learning was quantified by decomposing the learned weights $\mathbf{w}$ into two components: one proportional to $\mathbf{e}$ and a second orthogonal to $\mathbf{e}$, i.e.,

$$\mathbf{w} = |\mathbf{w}|(\mathbf{e} \cos \theta + \mathbf{e}_\perp \sin \theta) \quad (45)$$

where $\theta$ is the angle between $\mathbf{w}$ and $\mathbf{e}$, and $\mathbf{e}_\perp$ is the unit vector perpendicular to $e$ in the plane defined by $\mathbf{e}$ and $\mathbf{w}$. This allows us to write the decision variable $h$ in terms of signal and noise components as

$$h = |\mathbf{w}|((\mathbf{e} \cdot \mathbf{x}) \cos \theta + (\mathbf{e}_\perp \cdot \mathbf{x}) \sin \theta)$$
$$= \underbrace{|\mathbf{w}|(2t-1)\Delta \cos \theta}_{\text{signal}} + \underbrace{|\mathbf{w}|(\mathbf{e}_\perp \cdot \mathbf{x}) \sin \theta}_{\text{noise}} \quad (46)$$

where the difficulty $\Delta = |\mathbf{e} \cdot \mathbf{x}|$ is the distance between $\mathbf{x}$ and the decision boundary, and the $(2t - 1)$ term simply controls which side of the boundary $x$ is on. This implies that the precision $\beta$ is proportional to $\cot \theta$, with a constant of proportionality determined by the dimensionality of $\mathbf{x}$.

In the case where the observations $x$ are sampled from distributions that obey the central limit theorem, then the noise term is approximately Gaussian implying that the optimal error rate for training the Perceptron, $\text{ER}^* = 15.87\%$.

To test this prediction we simulated the Perceptron learning rule for a range of training error rates between 0.01 and 0.5 in steps of 0.01 (1000 simulations per error rate). Stimuli, $\mathbf{x}$, were 100 dimensional and independently sampled from a Gaussian distribution with mean 0 and variance 1. Similarly, the true weights $\mathbf{e}$ were sampled from a mean 0, variance 1 Gaussian. To mimic the effect of a modest degree of initial training, we initialized the weight vector $\mathbf{w}$ randomly with the constraint that $|\theta| < 1.6\pi$. The difficulty $\Delta$ was adjusted on a trial-by-trial basis according to

$$\Delta = F^{-1}(\text{ER}) \lambda \tan \theta \quad (47)$$

which ensures that the training error rate is clamped at ER. The degree of learning was captured by the precision $\beta$.

**Application to the two-layer neural network**. To implement the two-layer network, we built a sigmoidal neural network with one hidden layer (of 50 neurons) and one output neuron. The weights between the input layer and the hidden layer and between the hidden layer and output layer were trained using the standard Backpropagation algorithm.

In order to clamp the error rate during training we first had to rate the images according to their 'difficulty'. To this end, we trained a teacher network with the same basic architecture (i.e., 50 hidden units and 1 output unit) until its performance was near perfect (training error rate = 99.6% for the Parity Task and 99.4% for the Magnitude Task; test error rate = 97% for the Magnitude Task and 95.6% for the Parity Task). We then used the absolute value of the decision variable from this network, $|h^{\text{teacher}}|$ as a proxy for the true difficulty, $\Delta$—with larger values of $|h^{\text{teacher}}|$ indicating easier stimuli to classify.

Weights in the network were initialized randomly from a Gaussian distribution (mean 0, variance 1). To achieve a fixed error rate during training, on each trial, we selected a stimulus that was closest to a target difficulty, $h^{\text{target}}$. This target difficulty was adjusted based on the performance of the network during training—increasing if the network classified the stimulus incorrectly, and decreasing if the network classified the stimulus correctly. More specifically, the target difficulty was adjusted as

$$h^{\text{target}} \leftarrow h^{\text{target}} + D(A^{\text{target}} - A^{\text{av}}) \quad (48)$$

where $D$ is the step size ($=1$), $A^{\text{target}}$ is the target training accuracy and $A^{\text{av}}$ is the running average of the accuracy from the last 50 trials.

On each trial we selected the 'eligible' stimulus whose value of $h^{\text{teacher}}$ was closest to $h^{\text{target}}$. To ensure that a given stimulus was not selected too often during training, stimuli were only eligible to be chosen if they had not been used in the last 50 trials.

Each initial state of the network was trained on either the Parity or Magnitude Task at a fixed training error rate between 5 and 30% in steps of 5%. At the end of training performance was assessed on the whole test set. This process was repeated 1000 times, with a new set of initial random weights each time.

**Application to Law and Gold model**. The model of perceptual learning follows the exposition in Law and Gold[11]. To aid comparison with that paper we retain almost all of their notation, with the three exceptions being their $\beta$ parameter, which we rename as $B$ to avoid confusion with the precision, their $\phi_i$ parameter which we rename as $F_i$ to avoid confusion with the parameters of the learner, and their learning rate parameter $\alpha$ which we write as $\eta$.

Following Law and Gold[11], the average firing rate of an MT neuron, $i$, in response to a moving dot stimulus with direction $\theta$ and coherence COH is

$$m_i = T(k_i^0 + \text{COH}(k_i^n + (k_i^p - k_i^n) f(\theta | \Theta_i))) \quad (49)$$

where $T$ is the duration of the stimulus, $k_i^0$ is the response of neuron $i$ to a zero-motion coherence stimulus, $k_i^p$ is the response to a stimulus moving in the

preferred direction and $k_i^n$ is the response to a stimulus in the null direction. $f(\theta|\Theta_i)$ is the tuning curve of the neuron around its preferred direction $\Theta_i$

$$f(\theta|\Theta_i) = \exp\left(-\frac{(\theta - \Theta_i)^2}{2\sigma_\theta^2}\right) \qquad (50)$$

where $\sigma_\theta$ (=30 degrees) is the width of the tuning curve which is assumed to be identical for all neurons.

Neural activity on each trial was assumed to be noisily distributed around this mean firing rate. Specifically, the activity, $x_i$, of each neuron is given by a rectified (to ensure $x_i > 0$) sample from a Gaussian with mean $m_i$ and variance $v_i$

$$v_i = F_i m_i \qquad (51)$$

where $F_i$ is the Fano factor of the neuron.

Thus each MT neuron was characterized by five free parameters. These free parameters were sampled randomly for each neuron such that $\theta_i \sim U(-180, 180)$, $k_i^0 \sim U(0, 20)$, $k_i^p \sim U(0, 50)$, $k_i^n \sim U(-k_i^0, 0)$ and $F_i \sim U(1, 5)$. Note that $k_i^n$ is set between $-k_i^0$ and 0 to ensure that the minimum average firing rate never dips below zero. Each trial was defined by three task parameters: $T = 1$ s, $\Theta = \pm 90$ degrees and $COH$ which was adjusted based on performance to achieve a fixed error rate during training (see below). As in the original paper, the number of neurons was set to 7200 and the learning rate, $\eta$ was $10^{-7}$.

The predicted reward $E[r]$ was computed according to Eq. (20). In line with Law and Gold (Supplementary Fig. 2 in ref. [11]), the proportionality constant $B$ was computed using logistic regression on the accuracy and absolute value of the decision variable, $|h|$, from last $L$ trials, $L = \min(300, t)$.

In addition to the weight update rule (Eq. (21)), weights were normalized after each update to keep the sum of the squared weights, $\sum_i w_i^2 = w_{amp}$ a constant

(=0.02). While this normalization has only a small overall effect (see Supplementary Material in ref. [11]), we replicate this weight normalization here for consistency with the original model.

To initialize the network, the first 50 trials of the simulation had a fixed coherence COH = 0.9. After this initialization period, the coherence was adjusted according to the difference between the target accuracy, $A_{target}$, and actual accuracy in the last $L$ trials, $A_L$, where $L = \min(300, t)$. Specifically, the coherence on trial $t$ was set as

$$COH_t = \frac{1}{1 + \exp(-\Gamma_t)} \qquad (52)$$

where $\Gamma_t$ was adjusted according to

$$\Gamma_{t+1} = \Gamma_t + d\Gamma(A_{target} - A_L) \qquad (53)$$

and $d\Gamma$ was 0.1.

To estimate the post-training precision parameter, $\beta$, we simulated behavior of the trained network on a set of 20 logarithmically spaced coherences between $10^{-3}$ and 1. Behavior at each coherence was simulated 100 times and learning was disabled during this testing phase. The precision parameter, $\beta$, was estimated using logistic regression between accuracy on each trial (0 or 1) and coherence; i.e.,

$$ACC \sim \frac{1}{1 + \exp(-\beta \times COH)} \qquad (54)$$

## Data availability
Data sharing not applicable to this article as no datasets were generated or analysed during the current study.

## Code availability
All code is publicly available on GitHub at https://github.com/bobUA/EightyFivePercentRule

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

## Acknowledgements
This project was made possible through the support of a grant from the John Templeton Foundation to J.D.C., a Center of Biomedical Research Excellence grant P20GM103645

from the National Institute of General Medical Sciences to A.S., and National Institute on Aging grant R56 AG061888 to R.C.W. The opinions expressed in this publication are those of the authors and do not necessarily reflect the views of the funders.

## Author contributions

R.C.W., A.S., M.S., and J.D.C. developed the idea and wrote the paper. R.C.W. derived mathematical results and ran simulations.

## Competing interests

The authors declare no competing interests.
