## [Peer Review File · Nature Communications]

Reviewers' comments:

Reviewer #1 (Remarks to the Author):

The broad question the authors aim to address is at what difficulty level learning proceeds most swiftly.

Moving to an idealized setting of binary prediction, the authors consider the case of a score function that is equal to the target variable Δ plus a zero-mean gaussian noise term. The score function is thresholded at zero for the purpose of making a binary prediction. Therefore, the error probability of the resulting binary predictor is $F(-\Delta/\sigma)$, where σ is the standard deviation of the noise term, and F is the cumulative density function of the normal distribution.

In the setup of the manuscript, gradient descent can be used to directly optimize $F(-\Delta/\sigma)$ as an objective function. In other words, a gradient step reduces σ and shrinks the error probability. Since the objective has no stationary points (as a function of σ) in the positive real line $(0, \infty)$, gradient descent converges towards 0.

The authors then ask at what error rate (scaling of Δ), gradient descent makes the largest step. This happens when the derivative of the objective with respect to σ is maximized, corresponding to an error rate of $\sim 15\%$.

Some concerns:

- The model assumes that gradient descent directly minimizes the error probability under a gaussian noise model. In reality, gradient descent usually minimizes a loss function that typically does not directly correspond to this error probability.
- The aforementioned issue is relevant to the claimed Perceptron result. The authors assert that the 85% rule applies to Perceptron learning. That's not clear to me at all, since the Perceptron algorithm is equivalent to minimizing the so-called hinge loss, which does not correspond to the gaussian error probability that the analysis applies to.
- The authors neglect the relationship between the trainable model parameters (denoted ϕ) and the scaling of the noise (σ). Except in simple linear models, it's often highly non-trivial to understand how gradient updates to ϕ change the distribution of errors. Taking this into account might mean that a larger step size (obtained by optimizing Δ) isn't necessarily better.

Conclusion:

I like the question the paper asks. The proposed model is pleasingly simple and the 85% rule jumps right out of the setup. The manuscript is well written and the authors did a great job with the exposition in general.

To which extent this simple model corresponds to realistic learning tasks remains a bit of a leap of faith in my opinion.

It would've helped me greatly to see experiments on non-trivial learning problems, even if as simple as standard image classification tasks. Curriculum learning has been proposed in this context, e.g. [7], but hasn't been empirically successful over the years.

I would've liked to see a more nuanced discussion of cases where the 15% rule fails and why. I can think of several situations. Rather than asserting that 15% is the answer in general, it would've been more informative to characterize in what scenarios this is a good guide.

The use of the phrase "under fairly mild assumptions" in the introduction should probably be changed. The assumptions are extremely strong and apply to the simplest idealistic setting one could think of. It's possible that the 15% rule holds well beyond these assumptions, but there is no point in being unclear about the nature of the assumptions.

Reviewer #2 (Remarks to the Author):

This is a nice paper. The topic is very important with clear implications for learning in humans, animals, and machines. The result is surprising in its simplicity.

Most of my comments are relatively minor requests for clarification. I have one more substantial suggestion. I was disappointed to see the theory tested only on the simplest types of classifiers, e.g. single layer neural networks. The impact of the result will be much more substantial, especially in machine learning, if it is demonstrated using more complex models and classification problems. For instance, I would love to see a demonstration on image classification, e.g. discriminating between two digit classes drawn from MNIST.

Other suggestions and requests for clarification:

- I believe the analysis hinges on Δ being independent from the network parameters ϕ . This assumption wasn't stated explicitly in the text, but it would clarify your analysis.
- What are the assumptions on the stimulus x (and its distribution)? I could be mistaken, but currently the analysis assumes just a single x . This is appropriate for online learning, assuming that x is under active control by a teacher and chosen at each step to maintain the 85% rule (as reflected in its Δ). What does your analysis have to say about batch learning? Also, since your analysis concerns just one stimulus at a time, all instances of "gradient descent" in the paper should be replaced by "stochastic gradient descent."
- I don't believe the perceptron learning rule is strictly a stochastic gradient descent based rule, since the threshold activation function is not differentiable. The analysis cited in [20] refers to an artificial neuron with a logistic activation function.
- It's not clear how Figure 3 was created. How many learning trials were presented for each of the network simulations in Figure 3?
- The discussion states that the analysis applies to Boltzmann machines. It's unclear to me why it would, since Boltzmann machines are a generative model rather than a binary classifier.
- Could your analysis be extended to multi-class classification in future work? This is worth commenting on.
- The paper needs a more substantial reference than a personal communication to support the claim that "participant engagement is often maximized when performance is maintained around 85%." (pg. 8)

Reviewer #1 (Remarks to the Author):

The broad question the authors aim to address is at what difficulty level learning proceeds most swiftly.

Moving to an idealized setting of binary prediction, the authors consider the case of a score function that is equal to the target variable Δ plus a zero-mean gaussian noise term. The score function is thresholded at zero for the purpose of making a binary prediction. Therefore, the error probability of the resulting binary predictor is $F(-\Delta/\sigma)$, where σ is the standard deviation of the noise term, and F is the cumulative density function of the normal distribution.

In the setup of the manuscript, gradient descent can be used to directly optimize $F(-\Delta/\sigma)$ as an objective function. In other words, a gradient step reduces σ and shrinks the error probability. Since the objective has no stationary points (as a function of σ) in the positive real line $(0, \infty)$, gradient descent converges towards 0.

The authors then ask at what error rate (scaling of Δ), gradient descent makes the largest step. This happens when the derivative of the objective with respect to σ is maximized, corresponding to an error rate of $\sim 15\%$.

Some concerns:

- The model assumes that gradient descent directly minimizes the error probability under a gaussian noise model. In reality, gradient descent usually minimizes a loss function that typically does not directly correspond to this error probability.

This is a fair point - to make things tractable we made some fairly strong assumptions about the noise. Our numerical experiments suggest these are reasonable in simple cases, including the new example with the MNIST dataset (see below), but there is no guarantee they would hold in more complex cases. One way in which we can relax the assumptions was buried in the Methods section of the original paper. Here we consider a wider class of noise distributions (that are parameterized by a scale factor, β) to obtain a more general result, presented in Equation 26. This more general expression allows us to compute optimal training error rates for other distributions such as Laplacian noise and Cauchy noise, where the optimal training error rates change to about 18% and 25% respectively. We now refer to this analysis more explicitly in the Discussion (p. 11)

Of course, in these more complex situations, our assumptions may not always be met. For example, as shown in the Methods, relaxing the assumption that the

noise is Gaussian leads to changes in the optimal training accuracy: from 85% for Gaussian, to 82% for Laplacian noise, to 75% for Cauchy noise (Equations 31 in the Methods).

- The aforementioned issue is relevant to the claimed Perceptron result. The authors assert that the 85% rule applies to Perceptron learning. That's not clear to me at all, since the Perceptron algorithm is equivalent to minimizing the so-called hinge loss, which does not correspond to the gaussian error probability that the analysis applies to.

This is a fair point and we have changed the text as a result. While not strictly gradient descent itself, there is a sense in which the Perceptron rule can be related to it. In MacKay (2003; Chapter 39, p. 475), a Perceptron-like learning rule is derived from gradient descent for a neuron with a soft threshold (i.e. a softmax activation function) rather than a hard threshold as in the traditional Perceptron. While this soft-threshold model is not quite the Perceptron, it can be made arbitrarily close to the hard-threshold model by turning up the softmax gain (or, alternatively, by scaling up the inputs \mathbf{x}). To reflect this in the paper, we have changed our wording about the Perceptron learning rule to refer to it as “*closely related (albeit not identical) to*” a gradient descent rule rather than implying equivalence.

- The authors neglect the relationship between the trainable model parameters (denoted ϕ) and the scaling of the noise (σ). Except in simple linear models, it's often highly non-trivial to understand how gradient updates to ϕ change the distribution of errors. Taking this into account might mean that a larger step size (obtained by optimizing Δ) isn't necessarily better.

We now refer more explicitly to this assumption in the derivation right after equation 5

After Eq 5 (p. 5):

Of course, this analysis ignores the effect of changing ϕ on the form of the noise — instead assuming that it only changes the scale factor, β , an assumption that likely holds in the relatively simple cases we consider here, although whether it holds in more complex cases will be an important question for future work.

Conclusion:

I like the question the paper asks. The proposed model is pleasingly simple and the 85% rule jumps right out of the setup. The manuscript is well written and the authors did a great job with the exposition in general.

We thank the reviewer for the positive comments!

To which extent this simple model corresponds to realistic learning tasks remains a bit of a leap of faith in my opinion.

It would've helped me greatly to see experiments on non-trivial learning problems, even if as simple as standard image classification tasks. Curriculum learning has been proposed in this context, e.g. [7], but hasn't been empirically successful over the years.

To address this point we tested the effect of training accuracy on test accuracy in the MNIST database of handwritten digits. In particular we trained a sigmoidal network with one hidden layer with the backprop algorithm. In order to make a binary classification task from this data set, we had the network learn to classify either the parity (odd or even) or magnitude (high or low) of the numbers in the images. These tasks allowed us to use all the images in the database for training and also made for a more difficult learning task than simply discriminating between two numbers.

Full details of the simulations are in the revised manuscript in the Methods section and Matlab code for the simulations is available at <https://github.com/bobUA/EightyFivePercentRule>. The results of the simulations are shown below, where we plot test accuracy against training error rate for the Parity and Magnitude tasks (Figure 3). For the both tasks, the optimal training accuracy peaks at ER = 15%.

I would've liked to see a more nuanced discussion of cases where the 15% rule fails and why. I can think of several situations. Rather than asserting that 15% is the answer in general, it would've been more informative to characterize in what scenarios this is a good guide.

We have tried to tone down the rhetoric throughout the paper and have included the following in the discussion to address cases where we can prove the 85% rule does not hold (specifically for

different noise distributions), but also to highlight other cases such as batch learning and more than two categories.

Of course, in these more complex situations, our assumptions may not always be met. For example, as shown in the Methods, relaxing the assumption that the noise is Gaussian leads to changes in the optimal training accuracy: from 85% for Gaussian, to 82% for Laplacian noise, to 75% for Cauchy noise (Equation 31 in the Methods).

More generally, extensions to this work should consider how batch-based training changes the optimal accuracy, and how the Eighty Five Percent Rule changes when there are more than two categories. In batch learning, the optimal difficulty to select for the examples in each batch will likely depend on the rate of learning relative to the size of the batch. If learning is slow, then selecting examples in a batch that satisfy the 85% rule may work, but if learning is fast, then mixing in more difficult examples may be best. For multiple categories, it is likely possible to perform similar analyses, although the mapping between decision variable and categories will be more complex as will be the error rates which could be category specific (e.g. misclassifying category 1 as category 2 instead of category 3).

The use of the phrase "under fairly mild assumptions" in the introduction should probably be changed. The assumptions are extremely strong and apply to the simplest idealistic setting one could think of. It's possible that the 15% rule holds well beyond these assumptions, but there is no point in being unclear about the nature of the assumptions.

We have changed this to:

Under the assumption of a Gaussian noise process underlying the errors this optimal error rate is around 85% ...

Reviewer #2 (Remarks to the Author):

This is a nice paper. The topic is very important with clear implications for learning in humans, animals, and machines. The result is surprising in its simplicity.

We thank the reviewer for their positive comments!

Most of my comments are relatively minor requests for clarification. I have one more substantial suggestion. I was disappointed to see the theory tested only on the simplest types of classifiers, e.g. single layer neural networks. The impact of the result will be much more substantial,

especially in machine learning, if it is demonstrated using more complex models and classification problems. For instance, I would love to see a demonstration on image classification, e.g. discriminating between two digit classes drawn from MNIST.

We have implemented this suggestion with a two-layer network on the MNIST dataset. We considered two tasks, classifying the parity or magnitude of the numbers. We used these tasks because they were slightly more complex than distinguishing between any two pairs of numbers (e.g. 1 vs 2 is quite easy) and because it allowed us to use the whole dataset for training - which gives us more training examples at all levels of difficulty.

Full details of the simulations are in the revised manuscript in the Methods section and Matlab code for the simulations is available at <https://github.com/bobUA/EightyFivePercentRule>. The results of the simulations are shown below where we plot test accuracy against training error rate for the Parity and Magnitude Tasks. For the both tasks, the optimal training accuracy peaks at the optimal difficulty of 15%.

Other suggestions and requests for clarification:

- I believe the analysis hinges on Δ being independent from the network parameters ϕ . This assumption wasn't stated explicitly in the text, but it would clarify your analysis.

This is correct. Δ is the "true" decision variable and so, by definition, should be independent of the parameters ϕ . We have now clarified this in the text in the footnote on page 3

Note that Δ itself, as the 'true' decision variable, is independent of ϕ

- What are the assumptions on the stimulus x (and its distribution)? I could be mistaken, but currently the analysis assumes just a single x . This is appropriate for online learning, assuming that x is under active control by a teacher and chosen at each step to maintain the 85% rule (as reflected in its Δ).

Apologies for not being more clear in the text, this reflects our bias in coming at this from a human/animal perceptual learning angle where stimuli are always presented one at a time. Of course in machine learning this is not always the case! We have tried to be more clear in the text on page 5

If stimuli are presented one at a time (i.e. not batch learning), ...

What does your analysis have to say about batch learning?

This is an interesting question. If learning is slow relative to the batch size, such that error rate is approximately constant throughout the batch, then the analysis should carry over. This assumes that the stimuli in each batch are chosen so that they all have the same difficulty, around 85% correct (choosing an 85:15 mixture of very easy and very hard stimuli would also give 85% correct for each batch but no learning!). We have included a paragraph in the Discussion on this point.

... in batch learning, the optimal difficulty to select for the examples in each batch will likely depend on the rate of learning relative to the size of the batch. If learning is slow, then selecting examples in a batch that satisfy the 85% rule may work, but if learning is fast, then mixing in more difficult examples may be best.

Also, since your analysis concerns just one stimulus at a time, all instances of "gradient descent" in the paper should be replaced by "stochastic gradient descent."

Thanks for catching this! The change has been made throughout the paper.

- I don't believe the perceptron learning rule is strictly a stochastic gradient descent based rule, since the threshold activation function is not differentiable. The analysis cited in [20] refers to an artificial neuron with a logistic activation function.

Agreed. However, the logistic activation function can be made arbitrary close to the hard threshold function by upping the gain (or scaling up the inputs, x). To reflect this in the text we refer to the Perceptron learning rule as being "closely related (albeit not identical)" to gradient descent.

- It's not clear how Figure 3 was created. How many learning trials were presented for each of the network simulations in Figure 3?

We used up to 1000 learning trials in the Perceptron simulations. The y-axis in Figure 3A and the x-axis in 3B then reflect the additional trials of training up to 1000 extra trials. Details have been added to the Methods section.

To test this prediction we simulated the Perceptron learning rule for a range of training error rates between 0.01 and 0.5 in steps of 0.01 (1000 simulations per error rate, 1000 trials per simulation).

In addition we have uploaded the code for all examples to GitHub at

<https://github.com/bobUA/EightyFivePercentRule>

- The discussion states that the analysis applies to Boltzmann machines. It's unclear to me why it would, since Boltzmann machines are a generative model rather than a binary classifier.

Apologies for this error! We have removed this reference.

- Could your analysis be extended to multi-class classification in future work? This is worth commenting on.

Extending to multi-class classification would indeed be an important next step, and should be possible. There will be additional factors to consider in this case, such as how the decision variable maps onto the classes. For example, in one case we could imagine a one-dimensional decision variable that is mapped to many categories (e.g. wavelength of light to the name of the color), in another we could imagine mapping a D-dimensional decision variable onto D different categories. In addition, we may want to consider more than just the error rate and instead consider the type of error - e.g. mistaking category 2 for category 1 but not category 3. We have included a reference to this point in the Discussion:

For multiple categories, it is likely possible to perform similar analyses, although the mapping between decision variable and categories will be more complex, as will be the error rates, which could be category specific (e.g. misclassifying category 1 as category 2 instead of category 3).

- The paper needs a more substantial reference than a personal communication to support the claim that "participant engagement is often maximized when performance is maintained around 85%." (pg. 8)

Fair enough! We have removed the reference and toned down that particular sentence to (p. 11)

In Psychology and Cognitive Science, the Eighty Five Percent Rule accords with the informal intuition of many experimentalists that participant engagement is often maximized when tasks are neither too easy nor too hard.

In reality, not a lot of work has probed this question directly, the best evidence comes from the staircasing literature. A meta-analysis of 82 studies from this literature showed that most studies used staircases targeting 80-85% accuracy (Garcia-Perez, 1998, Table 1)

M A Garcia-Perez. Forced-choice staircases with fixed step sizes: asymptotic and small-sample properties. *Vision Res*, 38(12):1861–81, Jun 1998.

** See Nature Research's author and referees' website at www.nature.com/authors for information about policies, services and author benefits

This email has been sent through the Springer Nature Tracking System NY-610A-NPG&MTS

Confidentiality Statement:

This e-mail is confidential and subject to copyright. Any unauthorised use or disclosure of its contents is prohibited. If you have received this email in error please notify our Manuscript Tracking System Helpdesk team at <http://platformsupport.nature.com> . Details of the confidentiality and pre-publicity policy may be found here <http://www.nature.com/authors/policies/confidentiality.html> Privacy Policy | Update Profile

REVIEWERS' COMMENTS:

Reviewer #2 (Remarks to the Author):

I had a positive impression of the original manuscript, and the new version confirms my impression. The MNIST experiments substantially strengthen the paper and the applicability of the conclusions to machine learning tasks. I feel my comments have been adequately addressed.